# Interventional Treatment Strategies in Intrahepatic Cholangiocarcinoma and Perspectives for Combined Hepatocellular-Cholangiocarcinoma

**DOI:** 10.3390/cancers15092655

**Published:** 2023-05-08

**Authors:** Timo Alexander Auer, Federico Collettini, Laura Segger, Uwe Pelzer, Raphael Mohr, Felix Krenzien, Bernhard Gebauer, Dominik Geisel, Clarissa Hosse, Wenzel Schöning, Uli Fehrenbach

**Affiliations:** 1Department of Radiology, Charité—Universitätsmedizin Berlin, Augustenburger Platz 1, 13353 Berlin, Germany; 2Berlin Institute of Health (BIH), Anna-Louisa-Karsch-Straße 2, 10178 Berlin, Germany; 3Department of Hematology, Oncology and Cancer Immunology, Charité—Universitätsmedizin Berlin, Augustenburger Platz 1, 13353 Berlin, Germany; 4Department of Hepatology and Gastroenterology, Charité—Universitätsmedizin Berlin, Augustenburger Platz 1, 13353 Berlin, Germany; 5Department of Surgery—CVK/CCM, Charité—Universitätsmedizin Berlin, Augustenburger Platz 1, 13353 Berlin, Germany

**Keywords:** combined hepatocellular-cholangiocarcinoma, cholangiocarcinoma, locoregional therapies

## Abstract

**Simple Summary:**

Combined hepatocellular-cholangiocarcinoma (cHCC-CCA) is a rare form of liver cancer with features of both hepatocellular and biliary tract cancer. Locoregional tumor therapies are an integral part of the treatment strategies of hepatocellular carcinomas (HCC) and intrahepatic cholangiocarcinoma (iCCA). Due to the rarity of cHCC-CCA, no interventional therapies have been established. This review provides an overview of current radiologic interventions for CCA (excluding options for eCCA), to review and appraise the existing literature on the topic, and to provide an outlook on whether such interventions may have a role as treatment for cHCC-CCA in the future.

**Abstract:**

cHCC-CCA is an uncommon type of liver cancer that exhibits clinical and pathological characteristics of both hepatocellular carcinoma (HCC) and cholangiocarcinoma (CCA), which are the two main forms of primary liver cancer. The similarity to HCC and CCA makes therapeutical strategies challenging. The poor prognosis of CCA in general, as well as for cHCC-CCA, is mainly attributable to the fact that diagnosis is often at an advanced stage of disease. During the last decade, locoregional therapies usually performed by interventional radiologists and its established role in HCC treatment have gained an increasing role in CCA treatment as well. These comprise a wide range of options from tumor ablation procedures such as radiofrequency ablation (RFA), microwave ablation (MWA), computed tomography high-dose rate brachytherapy (CT-HDRBT), and cryoablation to transarterial chemoembolization (TACE), including the option of intra-arterial administration of radioactive spheres (transarterial radioembolization—TARE), and much attention has focused on the potential of individual concepts in recent years. The purpose of this review is to provide an overview of current radiologic interventions for CCA (excluding options for eCCA), to review and appraise the existing literature on the topic, and to provide an outlook on whether such interventions may have a role as treatment for cHCC-CCA in the future.

## 1. Introduction

Biliary tract cancers (BTCs) are a heterogeneous group of rare malignancies including intrahepatic cholangiocarcinoma (iCCA) and extrahepatic cholangiocarcinoma (eCCA). The latter are further divided into perihilar cholangiocarcinoma (pCCA) and distal cholangiocarcinoma (dCCA), gallbladder cancer (GBC), and cancers of the ampulla of Vater (AVC) [1,2,3,4]. Worldwide, cholangiocarcinoma (CCA), which arises from the ductular epithelium of the biliary tree, accounts for 3% of all gastrointestinal cancers [5]. CCA are rare but account for 10–15% of all primary liver cancers and represent the most common primary non-hepatocellular carcinoma (non-HCC) malignancy in the non-cirrhotic liver [4,5,6]. CCA is the second most common primary liver tumor while iCCA accounts for 10–20% of all CCAs, which has a dismal prognosis while early surgery remains the only curative treatment option [4,5,7,8,9]. Although the incidence of CCA varies geographically with a predominance in Asian countries, studies indicate that, especially in Western populations, its occurrence is linked to risk factors such as hepatic inflammation, fibrosis, and cirrhosis [10,11,12,13]. In addition to classic CCA and its variants, combined hepatocellular-cholangiocarcinoma (cHCC-CCA) exists as a rare form of primary liver cancer and accounts for less than 5% of the former [14]. The biphenotypic cHCC-CCA exhibits histological features of hepatic and biliary differentiation. Due to the rarity of the diagnosis, there is currently no guideline-based standard therapy for cHCC-CCA [4].

The poor prognosis of CCA in general, as well as for cHCC-CCA, is mainly attributable to the fact that diagnosis is often at an advanced stage of disease when only a small proportion of patients can still be treated with surgical resection. Approximately 70% of patients have unresectable or metastatic cancer at the time of diagnosis [3,15,16,17]. In these patients, systemic chemotherapy is considered the standard of care. Unfortunately, no drug therapies including the combination of PD-L1 plus chemotherapy provide a significant survival advantage [12,18,19].

During the last decade though, locoregional therapies usually performed by interventional radiologists and its established role in HCC treatment have gained an increasing role in CCA treatment as well [20]. These comprise a wide range of options from tumor ablation procedures such as radiofrequency ablation (RFA), microwave ablation (MWA), computed tomography high-dose rate brachytherapy (CT-HDRBT), and cryoablation to transarterial chemoembolization (TACE), including the option of intra-arterial administration of radioactive spheres (transarterial radioembolization—TARE), and much attention has focused on the potential of individual concepts in recent years. Due to the fact that CCAs are quite resistant to systemic chemotherapies and the fact that affected patients continue to have a poor prognosis and quality of life, interest has shifted to the transarterial treatment options—TACE and selective internal radiation therapy (SIRT). While all of the above-mentioned interventional procedures have the potential to reduce tumor volume, reliable data on neoadjuvant treatment strategies are sparse. The purpose of this review is to provide an overview of current radiologic interventions for CCA (excluding options for eCCA), to review and appraise the existing literature on the topic, and to provide an outlook on whether such interventions may have a role as treatment for cHCC-CCA in the future. 

## 2. Ablation Procedures

In general, ablative procedures can be distinguished in terms of their mode of action. A schematic overview is provided in the Figure 1 [21].

### 2.1. Thermal Ablation

While tumor ablation as such is well established for the curative treatment of early HCC, its role in iCCA is far less clear [22]. One reason for the former are surveillance programs for individuals with cirrhosis, improving the detection of early HCC when lesions are still small and amenable to successful ablation [22]. 

In principle, thermal ablation is possible and clinically effective in patients with iCCA lesions up to 3 cm in diameter. However, there is also evidence suggesting a benefit in lesions up to 5 cm [23,24]. Thermal ablation techniques such as RFA and microwave ablation (MWA) have a C2 recommendation as a “may” option for “small lesions that are not surgically accessible” in the EASL guidelines and are also mentioned as treatment for irresectable iCCA in the NCCN guidelines iCCA [19].

#### 2.1.1. Radiofrequency Ablation (RFA)

Among all ablation techniques, RFA is the most widely studied energy-based method. RFA utilizes high-frequency alternating electric currents that cause cell death by heating tissue through the generation of frictional heat by rapid electron vibration [24,25]. The effectiveness and heat generation of RFA are also highly dependent on the conductivity of the tissue in the target area, which in turn is significantly determined by its water content. Due to this mechanism, the ablation zone is limited in size as tissue adjacent to the electrode is warming up and consecutively desiccates, while acting as an “insulating sleeve” which degenerates the heat [24,25]. An efficacy-limiting factor of RFA is the fact that vessels adjacent to the ablation zone oppose temperature development, which results in incomplete ablation. This phenomenon is known as the “heat-sink effect” and can be counteracted by placing several needles [24,26,27].

Fu et al. (2012), who treated 26 iCCA lesions in 17 patients with RFA, reported 1-year, 3-year, and 5-year survival rates of 84.6%, 43.3%, and 28.9%, respectively, with an overall complication rate of 3.6% (1 out of 28 sessions) [23]. 

Han et al. (2015) conducted a meta-analysis with a comprehensive literature search to identify studies describing the use of RFA in the treatment of iCCA [28]. The purpose of this meta-analysis was to evaluate and review the clinical efficacy and safety of RFA in iCCA. The authors concentrated on data describing overall survival (OS), local tumor control (LTC) and complications, respectively. Seven observational studies including a total of 84 patients were reviewed. The pooled 1-year, 3-year, and 5-year survival rates were 82%, 47%, and 24%. One or two major complications occurred in four studies, and one patient died of a liver abscess and subsequent sepsis despite treatment with percutaneous drainage and antibiotics. The authors conclude that RFA is a locoregional treatment option that prolongs survival in patients with iCCA who are not candidates for surgery [28]. This is in line with another more recent systematic review and meta-analysis of thermal ablations in general in the treatment of iCCA conducted by Kim et al. (2022) [29]. The authors included 20 observational studies and reviewed 917 patients with both primary and recurrent iCCA. The pooled proportion of technical efficacy was 91.9%, while 1-, 3-, and 5-year OS rates were 82.4%, 42.1%, and 28.5%, respectively. Pooled 1- and 3-year recurrence-free survival (RFS) rates were 40.0% and 19.2%, respectively. The incidence of major complications was 5.7%. Furthermore, the authors identified tumor size (>3 cm), multiple tumors, and age (>65 years) as factors associated with shorter OS [29]. Kim et al. concluded that, comparable to its use in HCC, thermal ablation is a successful alternative to surgical resection with a good safety profile, especially for patients with a single iCCA smaller than 3 cm [29]. Of note, Kim et al. included both RFA and MWA in their analysis but excluded any kind of combined treatment [29]. Interestingly, long-term survival after thermal ablation procedures is comparable to that of surgical resection in a meta-analysis of Mavros et al. (2014), who reported median OS after curative resection of 28–30 months and a 5-year survival rate of about 30%, in line with the current literature (compared to 28.5% after thermal ablation in the study of Kim et al. (2022) [29,30,31,32]. At this point, it should be mentioned that patients who undergo locoregional therapies tend to have a worse general condition and therefore are not eligible for surgery [33]. 

#### 2.1.2. Microwave Ablation (MWA)

MWA has evolved over the years and has the potential to overcome some of the well-known limitations of RFA. Microwave technology induces frictional heating of biological tissues through electromagnetic radiation-induced rotation of dipole molecules, such as water, and can thus generate higher ablation temperatures than RFA [24,34,35]. Advantages of the higher tissue temperatures achieved with MWA include the creation of larger ablation zones in shorter heating times and a reduction of the heat-sink effect [35]. 

Since MWA is a rather new procedure, less data are available on its use in patients with iCCA. In 2018, Zhang et al. retrospectively investigated outcomes of MWA and explored prognostic factors for patient survival in a population of 107 patients with 171 iCCA lesions less than 5 cm in size [36]. The median follow-up after MWA was 20.1 months. Zhang et al. recorded no procedure-associated death and an overall procedure-associated major complication rate of 2.8%. Median progression-free survival (PFS) after MWA was 8.9 months; PFS rates after 6, 12, 18, and 24 months were 67.4%, 41.5%, 18.2%, and 8.7%, respectively. Median OS was 28.0 months; OS rates after 1, 3, and 5 years were 93.5%, 39.6%, and 7.9%, respectively. The authors concluded MWA to be safe and effective, while Child–Pugh class A disease and a smaller number of tumor lesions predicted longer PFS and OS in patients with iCCA treated by MWA [36]. These results are in line with those of Xu et al. (2019), who compared ultrasound-guided percutaneous MWA with surgical resection in 121 patients with recurrent iCCA [37]. In this study, 56 patients underwent MWA and 65 patients underwent surgical resection. Survival, recurrence, and liver function were comparable in the two groups. OS and RFS were comparable after MWA and surgical resection. The estimated 5-year OS rates were 23.7% after MWA and 21.8% after surgical resection; for RFS, the estimated 3-year rates were 33.1% after MWA and 30.6% after surgery. Notably, major complication rates were higher in the surgery subgroup than in the MWA subgroup (13.8% vs. 5.3%) [37]. 

Another parallel to HCC was provided by the results of Yang et al. (2015), who retrospectively evaluated the safety and efficacy of ultrasound-guided percutaneous MWA combined with simultaneous TACE in the treatment of 26 patients with advanced iCCA [38]. Complete ablation was accomplished in 92.3% of lesions, and no major complications were recorded. Median PFS and OS were 6.2 and 19.5 months, respectively. The 6-, 12-, and 24-month survival rates were 88.5%, 69.2%, and 61.5%, respectively [38]. In addition, some investigators performed both RFA and MWA in patients with iCCA, but they only reproduced the results already outlined above [39,40]. In conclusion, many basic considerations regarding hyperthermic ablation can be transferred from HCC to iCCA.

#### 2.1.3. Cryoablation

Cryoablation induces cell death by freezing processes affecting the cell membrane and organelle damage by dehydration and consecutive osmotic pressure changes resulting from the formation of intra- and extracellular ice crystals [24,41]. The needle tip is cooled down through a cooling mechanism based on the Joule–Thomson effect [41]. Basically, there are three main advantages of cryoablation [41]. First, the good visualization of the ice ball and the ablation zone during the procedure. Second, cryoablation has a lower risk of thermal damage compared with hyperthermal techniques as shown in Figure 2 [24,42,43]. 

A possible disadvantage of this technique is that it can be time-consuming and more expensive which is due to the time-consuming freezing cycles and the usage of gas [24,45,46]. A potential major complication is cryoshock, which was reported to occur in 0.3 to 2.0% of cases, especially when liver lesions were treated [47]. Overall, only few data are available on cryoablation in patients with iCCA, and published data were mainly obtained in studies on cryoablation in the liver with a small proportion of 3–6 patients having iCCA [24,45,46].

### 2.2. Non-Thermal Ablations

#### 2.2.1. Computed Tomography High-Dose-Rate Brachytherapy (CT-HDRBT)

Although hyperthermal procedures in general are the most frequently and most widely used ablation techniques in the clinical setting, they still have a few limitations, especially in terms of the number of lesions that can be treated and other factors such as the heat-sink effect that preclude thermal ablation in some lesion locations [48]. CT-HDRBT is a minimally invasive, radioablative technique that has shown promising results in the management of primary and secondary liver tumors, especially including tumors not readily accessible to thermal ablation [49] (Figure 3). 

In 2012, Schnapauff et al. investigated 15 patients with irresectable iCCA treated with CT-HDRBT. All patients were treated using an iridium-192 source introduced through CT-guided percutaneously placed catheters in an afterloading technique. After a median follow-up of 18 months after local ablation, six of the fifteen patients were still alive; four of them did not receive further chemotherapy and were regarded as complete response (CR). Median local tumor control after CT-HDRBT was 10 months and median OS was 14 months after CT-HDRBT and 21 months after initial diagnosis [50].

In the same year, Kamphues et al. published their preliminary experience with CT-HDRBT as an alternative treatment for hepatic recurrence of iCCA in ten patients [51]. The median OS of all patients after primary liver resection was 85 months with 1- and 5-year OS rates of 100% and 78.7%, respectively. After iCCA recurrence, a total of 15 CT-HDRBT procedures were performed, alone or combined with other treatments for recurrence, and no major complications occurred. The 1-year and 5-year survival rates after treatment of recurrent iCCA were 77.1% and 51.4%, respectively [51].

In 2018, Jonczyk et al. investigated and compared outcomes of CT-HDRBT in patients with iCCA up to 4 cm in size versus lesions larger than 4 cm. The authors included a total of 61 patients and performed 91 interventions. LTC was better in the group of smaller tumors up to 4 cm. Though interesting, there were no statistically significant differences in PFS and OS: in the subgroup of iCCA < 4 cm, PFS was five months and OS was fifteen and a half months. In the subgroup of iCCA > 4 cm, PFS was three months and OS was ten months. The authors concluded CT-HDRBT to be an adequate treatment in iCC even for lesions ≥ 4 cm as long as full coverage with therapeutic doses can be achieved [52]. 

Consistent with data on HCC, CT-HDRBT seems to be a highly promising ablation procedure for inconveniently located and large iCCA lesions. In addition to CT-HDRBT, there are also some data existing about the opportunity of intraluminal brachytherapy as a palliative treatment option for unresectable iCCA [53].

#### 2.2.2. Irreversible Electroporation (IRE)

IRE uses high electrical voltages to induce cell apoptosis throughout the destruction of cell membranes by electroporation [54]. Since IRE stimulates tumor cells in particular, while physiological structures are spared to a certain extent, its use can be considered at critical sites [24,55]. Data are low on IRE and iCCA while for cHCC-CCA no data exist. In a review from Tian el al. (2017), the authors evaluated the efficiency of IRE for hepatic malignant tumors and included a total of 300 patients. The pooled data indicated that IRE could be a minimal invasive and effective approach for patients who had preoperative poor liver function or those whose masses were in refractory locations where surgical resection was unsuitable, which could be potentially transferred to either iCCA as well as for cHCC-CCA [56].

Please find all discussed studies regarding ablational procedures on iCCA in Table 1.

## 3. Transarterial Procedures

Different transarterial procedures can be distinguished according to their mode of action. An overview is provided in Figure 4.

### 3.1. Transarterial Chemoembolization (TACE)

TACE is one the most commonly reported liver-directed therapy in iCCA patients, mainly because of the fact that TACE is well known as a standard treatment for patients with HCC [22,57]. TACE takes advantage of the fact that an arterial access, usually via the groin or the arm, can be used to pre-probe the liver with a catheter. From here, the tumor can be selectively probed and a mixture of an embolic agent and a chemotherapeutic agent can be administered [57]. Thus, TACE has two basic therapeutic effects. On the one hand, the embolic agent (usually lipiodol) causes occlusion of the tumor-feeding artery with subsequent hypoxia in the lesion [57]. As a further effect, selective administration allows use of significantly higher concentrations of the chemotherapeutic agent. Use of a mixture of lipiodol and chemotherapeutic agent is also known as cTACE (with the c meaning conventional [57]. In addition, it is possible to combine TACE with local ablative procedures as shown in Figure 5. 

Accordingly, it seems logical that TACE is especially suitable for treating arterially hypervascularized tumors [58,59]. Another important benefit of TACE is that local administration which reduces systemic toxicity is limited while hepatotoxicity can also be partially compensated due to the dual arterial and venous blood supply of the liver.

One of the largest studies, conducted by Park et al. (2011), showed a survival benefit of transarterial chemoembolization compared with supportive care alone in advanced iCCA patients: median survival was 12.2 months in the TACE group (n = 72) and 3.3 months in the control group (n = 83), although more than 50% of patients in both groups developed extrahepatic metastases [60]. Patients in the TACE group were treated with a cisplatin-based regime. In the TACE group, significant hematological and non-hematological toxicity events were observed in 13–24% of cases, while no patient died within 30 days after TACE [60]. No patient had a CR but the majority showed either a partial response (PR—23%) or stable disease (SD—66%). Progressive disease (PD) was only observed in 11% [60].

Kiefer et al. (2011) investigated TACE in iCCA and adenocarcinoma of unknown primary in 62 patients using a mixture of mitomycin C, doxorubicin, and cisplatin. The study showed an OS of 15 months. In this study, which was conducted at two centers, 165 TACE procedures were performed and major complications were only observed in 3% [61]. Note though that both the study of Park et al. (2011) and that of Kiefer et al. (2011) had retrospective designs without control groups [60,61]. 

Vogl et al. (2012) treated 115 patients with different protocols including mitomycin C alone, gemcitabine alone, gemcitabine and mitomycin C or gemcitabine, and mitomycin C and cisplatin [62]. Overall, 819 TACE procedures were performed as patients underwent repeat TACE. The median OS in this study was 13 months. Certainly, the most interesting thing about this study is the frequency with which Vogl et al. treated their patients. The patients were treated at 4-week intervals and the average number of treatments was 7.1 TACE procedures per patient. Of note, there were no documented major complications, which highlights the good tolerability of TACE. Nevertheless, the study was conducted and analyzed retrospectively, and no uniform protocol was used [62].

In addition to cTACE, which is certainly the most common technique apart from HCC, there are other TACE techniques, most notably TACE with drug-eluting beads (DEB-TACE). DEB-TACE has been commercially available on a large scale since 2006 and is used in numerous centers. Regarding its use in the treatment of HCC, there is an ongoing debate of its efficacy compared to cTACE [63,64]. DEB-TACE combines the release of drug-carrying beads with a consecutive reduction of blood flow by embolization.

Kuhlmann et al. (2012) compared cTACE vs. DEB-TACE for unresectable iCCA in a prospective study and retrospectively compared outcomes with those of patients treated by a combination of oxaliplatin and gemcitabine. Following predefined study protocols, 26 patients with histologically proven iCCA were treated with DEB-TACE (200 mg irinotecan) and 10 patients were treated with cTACE using 15 mg mitomycin C mixed with 5–10 mL of ionized oil (lipiodol), followed by embolization with gel foam. DEB-TACE resulted in PFS of 3.9 months and overall survival (OS) of 11.7 months, compared with a PFS of 1.8 months and OS of 5.7 months in patients treated with cTACE, and a PFS of 6.2 months and OS of 11.0 months in patients treated with oxaliplatin and gemcitabine [65]. Overall, the study demonstrates that treatment of iCCA with DEB-TACE is feasible and may be even superior to cTACE [65].

Despite the preliminary results just reviewed, it must be noted that transarterial procedures have not become established for the treatment of iCCA. There is a lack of prospective controlled or multicenter studies. A recently published study of Yang et al. (2022) though, evaluated the efficacy and safety of DEB-TACE combined with immune checkpoint inhibitors (ICI) in unresectable iCCA using a propensity score matching analysis [66]. The study included 49 patients with unresectable iCCA, 20 in the DEB-TACE&ICI subgroup and 29 in the chemotherapy subgroup. The patients in the DEB-TACE&ICI subgroup had a higher objective response rate (55.0% vs. 20.0%, *p* = 0.022), higher PFS (median, 7.2 vs. 5.7 months, *p* = 0.036), and higher OS (median, 13.2 vs. 7.6 months, *p* = 0.015) than those in the chemotherapy subgroup. Multivariate analysis suggested that chemotherapy, tumor size > 5 cm, and presence of multiple tumors were independent risk factors for poorer PFS and OS. The incidence of therapy-related adverse events was similar in the two groups [66]. The authors concluded that DEB-TACE plus ICI improved survival and was well tolerated by patients with unresectable iCCA. Locoregional therapies in combination with immunotherapies are currently being discussed. Preliminary results suggest that transarterial therapies could have an added value in this setting [66]. 

In addition to transarterial procedures aimed at permanent occlusion (cTACE and DEB-TACE), there are techniques for temporary occlusion of feeding vessels with degradable starch microspheres (DSM-TACE) [67]. DSMs were developed to temporarily occlude feeding vessels for up to 80 min [68]. While promising data exist for advanced HCC, experience with DSM-TACE in iCCA is still insufficient [67]. 

### 3.2. Transarterial Radioembolization (TARE)

Transarterial radioembolization (TARE) arose as a treatment option for iCCA [69,70,71]. Although several possible emitters are available, yttrium-90 (^90^Y) microspheres have been the most widely validated and have been classified as effective for liver tumors such as HCC or hepatic metastasis [69,70]. ^90^Y is a pure beta-emitter with an average energy of 0.94 megaelectron volt (half-life of 2.67 days) and decays to zirconium-90 [72,73,74]. In its active state, ^90^Y has a tissue penetration depth of 2.5 mm and a maximum range of 11 mm. Similar to chemoembolic agents in TACE, the microspheres reach the tumor via arterial feeders [72,73,74]. Optimally, most of the tumor tissue is irradiated locally without causing too much damage to the surrounding liver. A unique feature of TARE is that it is often performed in a two- or even three-step procedure. First, the patient is evaluated in a trial run. Treatment proper takes place 2–3 weeks later and can be carried out as whole or partial liver radioembolization [72] (Figure 6).

As early as 2012, Hoffmann et al. published a study investigating ^90^Y-TARE in 33 patients with unresectable iCCA [70]. Although the study population was heterogenous, Hoffmann et al. reported PR rates after three months of up to 36.4%, while 51.1% of patients remained stable. Median OS was 22 months in this population, and no relevant adverse events or higher-grade toxicities occurred. Although the study population was small and the analysis retrospective, the authors concluded TARE to be a promising treatment option for patients with unresectable iCCA [70].

Gangi et al. (2018) evaluate the efficacy and safety of ^90^Y-TARE in 85 consecutive patients with unresectable iCCA. Median OS from diagnosis was 21.4 months, while median OS from ^90^Y-TARE was 12.0 months. At 3 months, 6.2% of patients had PR, 64.2% had SD, and 29.6% had PD. The median OS from ^90^Y-TARE was significantly longer in patients with Eastern Cooperative Oncology Group (ECOG) scores of 0 or 1 than in patients with an ECOG score of 2 (18.5 vs. 5.5 months, *p* = 0.0012). The authors conclude that TARE has a good efficacy and an acceptable safety profile in the treatment of unresectable iCCA [69].

Schaarschmidt et al. (2022) investigated ^90^Y-TARE in iCCA in a retrospective, but multicentric setting, including a total of 138 radioembolizations performed in 128 patients of five tertiary care centers [75]. The purpose of this study was to identify factors associated with an improved median OS. ^90^Y-TARE was performed as first-line treatment in 25.4%, as second-line treatment in 38.4%, and as salvage treatment in 36.2%. In patients receiving first-line, second-line, and salvage radioembolization, the disease control rate was 68.6%, 52.8%, and 54.0% after 3 months; 31.4%, 15.1%, 12.0% after 6 months, and 17.1%, 5.7%, and 6.0% after one year, respectively [75]. In patients with ^90^Y-TARE as first-line, second-line, and salvage treatment, the median OS was 12.0 months, 11.8 months, and 8.4 months, respectively. No significant differences between the three groups were observed and, interestingly, the hepatic tumor burden did not significantly influence the median OS. Schaarschmidt et al. concluded that, especially in patients with advanced iCCA, second-line and salvage TARE might be important treatment options [75]. 

Schartz et al. (2022) investigated the overall efficacy and survival profile of ^90^Y-TARE for unresectable intrahepatic cholangiocarcinoma (ICC). In this systematic literature review, the colleagues included 21 studies with a total of 921 patients. The overall rate of disease control was 82.3%. In 11% of the cases, patients were downstaged to being surgically resectable. From the time of radioembolization, PFS was 7.8 months and median OS was 12.7 months. The authors concluded that ^90^Y-TARE for unresectable ICC results in substantial downstaging, disease control, and survival [76].

Overall, the studies presented clearly show that TARE is a treatment option for patients with advanced iCCA. Nevertheless, three aspects need to be discussed with regard to an outlook for the future of TARE in iCCA. First, the research on radiospheres is far from complete and the use of other radioembolization agents could also affect outcomes in the treatment of iCCA [69]. Second, personalized dosimetry in TARE and the possibility to adjust and increase dosage levels in liver compartments offers unique treatment options compared to other locoregional therapies [77]. Along the same lines, the term radiation segmentectomy has been coined in recent years. Radiation segmentectomy refers to the (super-)selective delivery of high radiation doses to the hepatic segment harboring the tumor [78]. Especially in HCC, this has led to a shift in indications. While TARE was only performed in advanced multifocal HCC before, its main indication nowadays is radiation segmentectomy for large solitary tumors [78,79]. The indications range from curative treatment, bridging to transplant procedures, and neoadjuvant therapy with the potential of hypertrophy induction in the non-diseased future liver remnant [78,79]. Although not everything shown for HCC is directly transferrable to iCCA, there are parallels that could also create new treatment options for cHCC-CCA. 

Please find all discussed studies regarding transarterial procedures on iCCA in Table 2.

## 4. Other Emerging Treatments

Another exciting emerging field is to use locoregional therapies in combination with immunotherapies. The latter have only a limited effect in the liver, which is due to the fact that the liver itself is an immunosuppressive organ and that tumors have a certain anti-immunological competence. Locoregional therapies can counteract these immunosuppressive mechanisms and thus enhance the efficacy of immunotherapies. Such an approach could be of particular interest for iCCA, in which all chemotherapeutic and immunological options have so far been unsuccessful.

## 5. Conclusions

In summary, all current treatment guidelines unanimously recommend resection of iCCA and cHCC-CCA whenever possible. However, most patients already have an unrespectable tumor burden at the time of diagnosis. In the absence of extrahepatic tumor manifestations, locoregional therapy may be considered. The available methods can be used both for tumor control and for neoadjuvant downstaging before surgery in iCCA. Unfortunately, no standardized approach has yet been established. This is primarily due to the often heterogeneous patient populations studied, the lack of prospective data, and the fact that no broadly accepted iCCA staging system comparable to the Barcelona Clinic Liver Cancer (BCLC) staging system used for HCC is available. However, it should be noted that the many retrospective data give reason to believe that locoregional therapy in iCCA appears to be effective and safe. In addition, many parallels to the treatment recommendations for HCC may be derived to iCCA from the BCLC criteria. The outlook for the future can only be addressed by familiar solutions such as randomized controlled trials and multicentric study designs. Since the data on interventional treatment of CCA has improved significantly in recent years and is already established in the treatment of HCC, local interventional therapy options seem promising for cHCC-CCA as well. In the same breath, given the range of techniques available, patient selection will play an increasingly important role in order to choose the best locoregional therapy for a given patient in terms of a precision oncology approach—a process that will shape interventional radiology and is far from over.

## Figures and Tables

**Figure 1 cancers-15-02655-f001:**
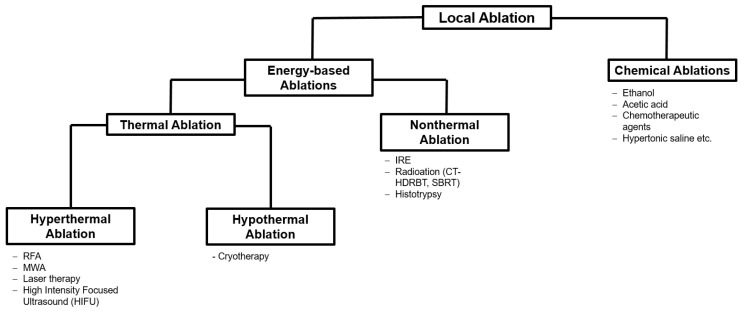
Schematic overview of types of local ablation therapies [21].

**Figure 2 cancers-15-02655-f002:**
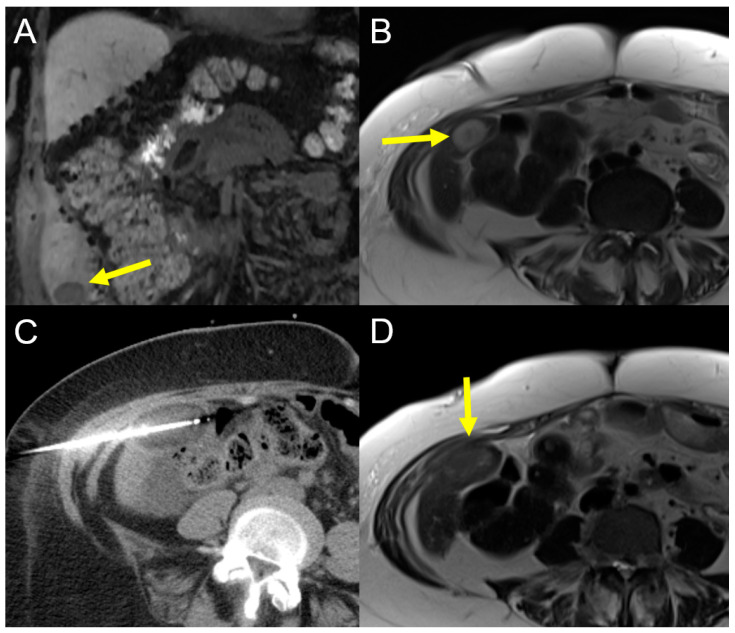
A 65-year-old female patient with recurrent iCCA (yellow arrow) at the caudate margin of segment V contiguous to the colon (**A**,**B**). (**C**) shows the second of two freezing cycles and one of two cryoablation needles placed inside the tumor (yellow arrow). The hypodense oval structure is the ice ball. (**D**) shows the control MRI after 4 weeks demonstrating a complete ablation zone with non-vital tissue and no signs of injury to the colon (yellow arrow). Third, cryoablation is associated with less intraprocedural pain [24,44].

**Figure 3 cancers-15-02655-f003:**
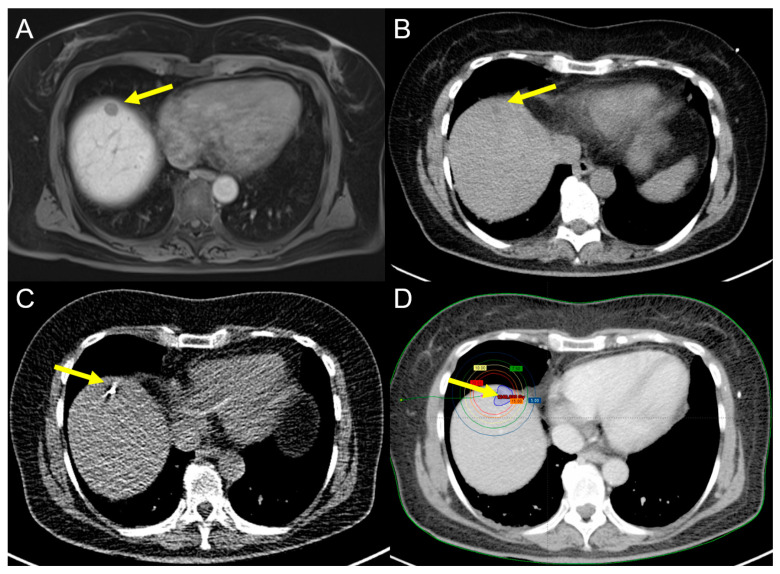
A 55-year-old female patient with recurrent iCCA (yellow arrow) at the resection margin (**A**). (**B**) is a CT scan obtained before CT-HDRBT catheter placement (yellow arrow pointing at the lesion). (**C**) shows the needle while placing the catheter (yellow arrow) under fluoroscopic guidance. (**D**) shows the treatment planning scan (yellow arrow) for brachytherapy in an afterloading technique.

**Figure 4 cancers-15-02655-f004:**
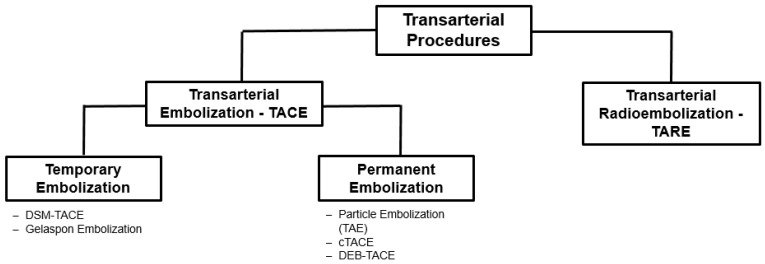
Schematic overview of transarterial embolization techniques.

**Figure 5 cancers-15-02655-f005:**
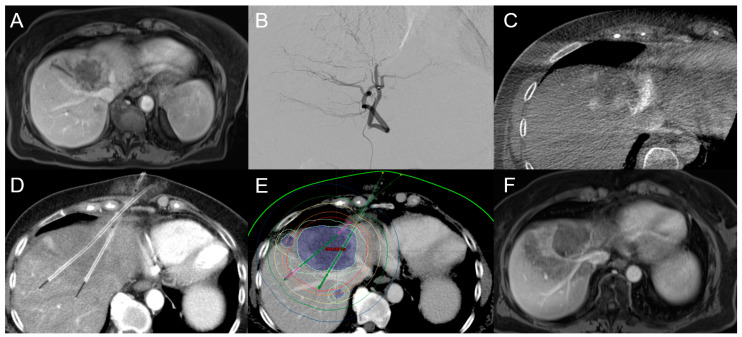
An 84-year-old female patient with initial diagnosis of unresectable iCCA with a diameter of up to 6.5 cm in segment IVa close to the inferior caval vein confluence (**A**). (**B**) shows the embolization position before TACE and (**C**) shows the peri-interventional cone beam CT scan. After TACE, CT-HDRBT was performed (the colored lines represent the isodose boundaries) (**D**,**E**), while (**F**) shows the huge ablation zone covering most of the iCCA mass.

**Figure 6 cancers-15-02655-f006:**
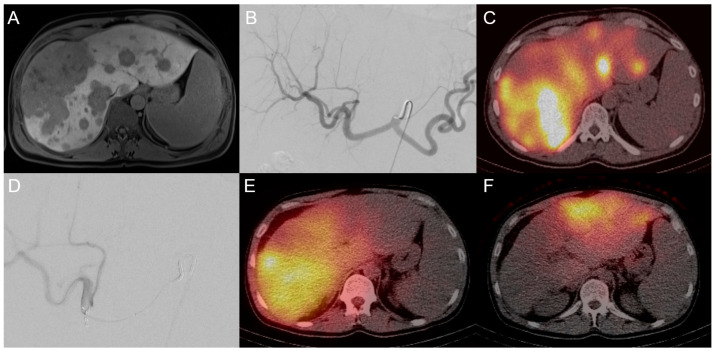
A 42-year-old male patient with initial diagnosis of unresectable ICCA with bilobar satellite metastases (**A**) who was referred to undergo evaluation for TARE (**B**). (**C**) is a post-evaluation single-photon emission computed tomography (SPECT) showing tracer accumulation in tumor lesions. Two weeks later, TARE of the right liver lobe was performed. (**D**) is the pretherapeutic embolization angiogram. (**E**) shows the post-therapeutic SPECT. After another eight weeks, TARE of the left liver lobe was performed with a good distribution of ^90^Y-spheres in the post-therapeutic SPECT (**F**).

**Table 1 cancers-15-02655-t001:** Overview of the cited studies regarding the ablational procedures on iCCA.

Procedures	Authors	Design	Endpoints *	Outcome
**RFA/MWA**				
**RFA**	Fu et al. (2012) [23]	Monocentric; Retrospective; n = 17	Tumor necrosis	96.2%
			OS (median)	33 months
			RFS (median)	17 months
			1Y-, 3Y-, 5Y-OS	84.5%; 43.3%; 28.9%
**RFA**	Han et al. (2015) [28]	Meta-Analyses; Systematic Review; n = 84	1Y-, 3Y-, 5Y-OS	82%; 47%; 24% (pooled)
			Major complications	2.3%
**RFA/MWA**	Kim et al. (2022) [29]	Meta-Analyses; Systematic Review; n = 917	Technical efficacy	91.9%
			1Y-, 3Y-, 5Y-OS	82.4%; 41.1%; 28.5%
			1Y-, 3Y-RFS	40%; 19.2%
			1Y-, 3Y-, 5Y-TTLTP	79.3%; 59.5%; 58.2%
			Major complications	5.7%
**MWA**	Zhang at al. (2018) [36]	Monocentric; Retrospective; n = 107	OS (median)	28.0 months
			1Y-, 3Y-, 5Y-OS	93.5%; 39.6%; 7.9%
			6 m-, 12 m-, 18 m-, 24 m-PFS	67.4%; 41.5%; 18.2%; 8.7%
			Major complications	2.8%
**MWA vs. SR**	Xu et al. (2019) [37]	Monocentric; Retrospective; n = 121	MWA:	
			5Y-OS (estimated)	23.7%
			3Y-RFS (estimated)	33.1%
			Major complications	5.3% (sign.)
			Surgical resection:	
			5Y-OS (estimated)	21.8%
			3Y-RFS (estimated)	30.6%
			Major Complications	13.8% (sign.)
**MWA + TACE**	Yang et al. (2015) [38]	Monocentric; Retrospective; n = 26	OS (median)	19.2 months
			PFS (median)	6.2 months
			6 m-, 12 m-, 24 m-OS	88.5%; 69.2%; 61.5%
			Major complications	0%
**Cryoablation**				
	Currently no significant evidence available			
**CT-HDRBT**				
**CT-HDRBT**	Schnaauff et al. (2012) [50]	Monocentric; Retrospective; n = 15	OS (median)	14 months
			LTC (median)	11 months
**CT-HDRBT**	Kamphues et al. (2012) [51]	Monocentric; Retrospective; n = 10	1Y-, 5Y-OS	77.1%; 51.4%
			Major complications	0%
**CT-HDRBT**	Jonczyk et al. (2018) [52]	Monocentric; Retrospective; n = 61	iCCA < 4 cm:	
			OS (median)	15.5 months
			LTC (median)	8.0 months (sign.)
			PFS (median)	5.0 months
			iCCA > 4 cm:	
			OS (median)	10.0 months
			LTC (median)	6.0 months (sign.)
			PFS (median)	3.0 months
**IRE**				
	Currently no significant evidence available			

* Survival times from interventional procedures out were selected in each case.

**Table 2 cancers-15-02655-t002:** Overview of the cited studies regarding the transarterial procedures on iCCA.

Procedures	Authors	Design	Endpoints *	Outcome
**TACE**				
**cTACE vs. Supportive Care**	Park et al. (2011) [60]	Monocentric; Retrospective; n = 155	cTACE:	
			OS (median (estimated))	12.2 months (sign.)

			Supportive care:	
			OS (median (estimated))	2.3 months (sign.)
**cTACE**	Kiefer et al. (2011) [61]	Bicentric; Retrospective; n = 62	OS (median)	20 months
			1Y-, 2Y-, 3Y-OS	75%; 39%; 17%
			PFS (median)	8 months
			1Y-PFS	28%
**cTACE**	Vogl et al. (2012) [62]	Monocentric; Retrospective; n = 155	OS (mean)	20.8 months
			OS (median)	13.0 months
			1Y-, 2Y-, 3Y-OS	52%; 29%; 10%
**cTACE vs.** **DEB-TACE vs. CTx**	Kuhlmann at al. (2012) [65]	Monocentric; Retrospective; n = 67	cTACE:	
			OS	5.7 months
			PFS	1.8 months

			DEB-TACE:	
			OS	11.7 months
			PFS	3.9 months

			CTx:	
			OS:	11.0 months
			PFS:	6.2 months
**DEB-TACE + ICI vs. CTx**	Yang et al. (2022) [66]	Monocentric; Retrospective; n = 49	DEB-TACE + ICI:	
			OS (median)	13.2 months (sign.)
			PFS (median)	7.2 months (sign.)
			Objective response rate	55.0% (sign.)

			Chemotherapy:	
			OS (median)	7.6 months (sign.)
			PFS (median)	5.7 months (sign.)
			Objective response rate	20.0% (sign.)
**DSM-TACE**				
	Currently no significant evidence available			
**TARE**				
**TARE**	Hoffmann et al. (2012) [70]	Monocentric; Retrospective; n = 33	OS (median)	22 months
			TTP (median)	9.0 months
	Gangi et al. (2018) [69]	Monocentric; Retrospective; n = 85	OS (median)	12.0 months
	Schaarschmidt et al. (2022) [75]	Multicentric; Retrospective; n = 128	1st-line Treatment:	
			OS (median)	12 months
			3 m-, 6 m-, 12 m-disease control	68.5%; 31.4%; 17.1%

			2nd-line Treatment:	
			OS (median)	11.8 months
			3 m-, 6 m-, 12 m-disease control	52.8%; 15.1%; 5.7%

			Salvage Treatment:	
			OS (median)	8.4 months
			3 m-, 6 m-, 12 m-disease control	54.0%; 12.0%; 6.0%
	Schartz et al. (2022) [76]	Meta-Analyses; Systematic Review; n = 921	OS (median)	12.7 months
			3 m-, 6 m-, 12 m-, 18 m-, 24 m-, 30 m-, 36 m-OS	84%; 69%; 47%; 31%; 30%; 21%; 5%
			PFS (median)	7.8 months
			Disease control rate	82.3%

* Survival times from interventional procedures out were selected in each case.

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
