# Peer review of "Interventional Treatment Strategies in Intrahepatic Cholangiocarcinoma and Perspectives for Combined Hepatocellular-Cholangiocarcinoma"

_cancers, 2023, doi:10.3390/cancers15092655_

Round 1
Reviewer 1 Report
Auer et al. provide a brief overview of current therapies for unresectable cholangiocarcinoma. Below are some suggestions and corrections to improve the manuscript.
The authors should also include up-to-date information regarding alternative methods (e.g., photodynamic therapy/PDT, irreversible electroporation/IRE) and combined hepatocellular cholangiocarcinoma treatments.
Consider to mention the following references: PMID 32258519, PMID 33723125, PMID 34283065, PMID 35116111, PMID 35219834.
Modify headings and subheadings:
2. Ablation Procedures
2.1. Thermal Ablation: RFA, MWA, PDT, Cryoablation
2.2. Nonthermal Ablation: CT-HDRBT, IRE
3. Transarterial Procedures
3.1. TACE
3.2. TARE
Line 56; here the authors describe also nonradiologic interventions
Lines 243-244; review the sentence
Line 326; …right liver lobe
Typo & text errors:
Line 218; …(Fig. 4)
Line 220; …embolization
Author Response
Reviewer #1:
C1: Auer et al. provide a brief overview of current therapies for unresectable cholangiocarcinoma. Below are some suggestions and corrections to improve the manuscript.
A: Thank you. Please find our amendments below.
C2: The authors should also include up-to-date information regarding alternative methods (e.g., photodynamic therapy/PDT, irreversible electroporation/IRE) and combined hepatocellular cholangiocarcinoma treatments.
C3: Consider to mention the following references: PMID 32258519, PMID 33723125, PMID 34283065, PMID 35116111, PMID 35219834.
A: Thank you for these impulses. Although not implemented we agree that IRE seems to be of emerging scope in iCCA treatment. We added a paragraph handling IRE in iCCA and tried to implement the recommended references. As we believe PDT is not an ideal procedure that can be used for solid hepatic tumors like cHCC-CCA, we did not included it.
C4: Modify headings and subheadings:
- Ablation Procedures
2.1. Thermal Ablation: RFA, MWA, PDT, Cryoablation
2.2. Nonthermal Ablation: CT-HDRBT, IRE
- Transarterial Procedures
3.1. TACE
3.2. TARE
A: We restructured the ablation procedures according to your advice.
C4: Line 56; here the authors describe also nonradiologic interventions
A: we mention this form of treatment here in the context of completeness.
C5: Lines 243-244; review the sentence
A: We did.
C6 Line 326; …right liver lobe
A: We corrected.
Typo & text errors:
Line 218; … (Fig. 4)
Line 220; …embolization
A: We corrected both typo & text errors.
Reviewer 2 Report
Comments to the Authors
This is a review article that discusses the current radiologic intervention options for cholangiocarcinoma (CCA) and its purpose is to provide an outlook whether such interventions may have a role as treatment for combined hepatocellular-cholangiocarcinoma (cHCC-CCA) in the future, as no standardized approach has been established yet. There are concerns that need to be addressed prior to consideration of publication as the main contribution of the paper is evaluating the different alternatives regarding radiologic interventions for CCA, leaving aside perspectives for cHCC-CCA.
Major Concerns
- The title of the article includes “perspectives for cHCC-CCA”, nevertheless it only describes therapeutic options for treatment of CCA and fails to explore treatment for cHCC-CCA. It must have deeper discussion in relation to recent findings, new developments or proposals regarding cHCC-CCA, hopefully in every treatment option (radiofrequency ablation [RFA], microwave ablation [MWA], computed tomography high-dose rate brachytherapy [CT-HDRBT], cryoablation and transarterial procedures) or instead the title should be re-written.
- In the introduction, biliary tract cancers (BTCs) are correctly classified, but at some point the authors should mention that the article will only deal or focus with treatment of intrahepatic cholangiocarcinoma (iCCA) as it does not mention extrahepatic cholangiocarcinomas (eCCA) anywhere in the article.
- The authors should revise the order in which the information is presented to improve readability. If the authors first mentioned thermal ablation by writing about hyperthermal techniques, then, they should refer to cryoablation to continue in line with thermal treatment options. Later on, they should discuss CT-HDRBT. If this suggestion is followed, consider changing the fist introductory sentence of the CT-HDRBT paragraph so that there is coherence between the paragraphs following this change.
- In the conclusion, the authors present a new idea that is not previously mentioned in the text regarding locoregional therapies in combination with immunotherapies (from line 386 - 392). Throughout these sentences a promising treatment alternative is explored, consider adding another subtitle, for example “other emerging treatments” to add this information and erase it from the conclusion.
Minor Concerns
- Line 21 please write “perihilar cholangiocarcinoma” to clarify the abbreviation “pCCA”.
- Line 36 please add commas in the sentence “The poor prognosis of CCA in general, as well as for cHCC-CCA, is mainly attributable to the fact that..”
- Line 44 please write the non-abbreviated form of hepatocellular carcinoma and then put it in the brackets: “hepatocellular carcinoma (HCC)”, as it is not previously mentioned.
- Line 52, is the word “poor” (or similar) missing next to prognosis? Please check.
- Line 71, 72, 199, 210, 229, 231, 314, 315, 367 please add reference at the end of the sentence.
- Line 72 RFA is already explained, please just mention “RFA”.
- Line 73 MWA is already explained, please just mention MWA.
- Line 100, please add to the end of the sentence, “...respectively.”
- Line 157, it is preferable if the headings are not with abbreviations. Please write heading as “computed tomography high-dose rate brachytherapy (CT-HDRBT)”
- Line 161-162, for the term “computed tomography high-dose rate brachytherapy” please only write the abbreviation “CT-HDRBT”.
- Line 167 there is a spelling error, please change “recection” for “resection”.
- Line 175 please write the full form of “CR”.
- In line 176, it is stated that the “median local tumor control was 10 months”, this was after CT-HDRBT? Explicit please.
- When referring to the number of patients, do so with a number or the number written in words. I suggest doing the numbers, as the majority of the paper is written this way. In line 179 please change “ten” for “10”.
- Paragraph from line 209-215 should be added to the previous one, as the authors are naming three advantages together.
- Line 221 it is preferable if the headings are not with abbreviations. Please write as heading as “Transarterial Embolization (TACE)”
- Line 224 please change “hepatocellular carcinoma” for “HCC” because it is already abbreviated in line 44.
- In line 232 there is a grammatical error, close the parenthesis before the reference.
- It is suggested that authors use a standard way in mentioning figures. Choose either “Fig number” or (Fig. number), you have used both in the article.
- Line 236, please add a space in between the number and the cm in “6.5cm”.
- Line 239, eliminate one point after the final sentence, you have two.
- Please remove sentence from line 240 to 243, the section about TACE is extensive and that sentence does not contribute on a large scale.
- Please rewrite sentence from lines 243-245.
- Line 352, eliminate a coma (there are two).
- Line 253 please only write the abbreviation “CR”, because it is previously non-abbreviated in line 175.
- Line 309 please write “Transarterial radioembolization (TARE)”, as it is preferable if the headings are not with abbreviations.
- Line 310, please write only the abbreviation “TARE”.
- In line 323 and 342, please change “ICCA” to “iCCA”.
- Line 360, please change the red dot for a black dot.
Author Response
Reviewer #2:
This is a review article that discusses the current radiologic intervention options for cholangiocarcinoma (CCA) and its purpose is to provide an outlook whether such interventions may have a role as treatment for combined hepatocellular-cholangiocarcinoma (cHCC-CCA) in the future, as no standardized approach has been established yet. There are concerns that need to be addressed prior to consideration of publication as the main contribution of the paper is evaluating the different alternatives regarding radiologic interventions for CCA, leaving aside perspectives for cHCC-CCA.
Major Concerns
- The title of the article includes “perspectives for cHCC-CCA”, nevertheless it only describes therapeutic options for treatment of CCA and fails to explore treatment for cHCC-CCA. It must have deeper discussion in relation to recent findings, new developments or proposals regarding cHCC-CCA, hopefully in every treatment option (radiofrequency ablation [RFA], microwave ablation [MWA], computed tomography high-dose rate brachytherapy [CT-HDRBT], cryoablation and transarterial procedures) or instead the title should be re-written.
A: Thank you very much for this critical and constructive comment. Furthermore, we would like to thank the reviewer for the detailed comments, which have improved the manuscript significantly. Regarding the first comment, we would like to stick to our writing style. As there is no standard of care for cHCC-CCA, all options so far are directed towards iCCA. The aim of the review was to elaborate and emphasize that in case of suspected cHCC-CCA, in the absence of surgical resection, the same locoregional therapy options as for iCCA should be considered.
Furthermore, according to Comment #1 of Reviewer #3 we changed the title.
- In the introduction, biliary tract cancers (BTCs) are correctly classified, but at some point the authors should mention that the article will only deal or focus with treatment of intrahepatic cholangiocarcinoma (iCCA) as it does not mention extrahepatic cholangiocarcinomas (eCCA) anywhere in the article.
A: We corrected.
- The authors should revise the order in which the information is presented to improve readability. If the authors first mentioned thermal ablation by writing about hyperthermal techniques, then, they should refer to cryoablation to continue in line with thermal treatment options. Later on, they should discuss CT-HDRBT. If this suggestion is followed, consider changing the fist introductory sentence of the CT-HDRBT paragraph so that there is coherence between the paragraphs following this change.
A: Thank you for this comment. According to Comment #4 of Reviewer #1 we changed the order.
- In the conclusion, the authors present a new idea that is not previously mentioned in the text regarding locoregional therapies in combination with immunotherapies (from line 386 - 392). Throughout these sentences a promising treatment alternative is explored, consider adding another subtitle, for example “other emerging treatments” to add this information and erase it from the conclusion.
A: We changed according to your advice.
Minor Concerns
- Line 21 please write “perihilar cholangiocarcinoma” to clarify the abbreviation “pCCA”.
A: We clarified.
- Line 36 please add commas in the sentence “The poor prognosis of CCA in general, as well as for cHCC-CCA, is mainly attributable to the fact that..”
A: We did.
- Line 44 please write the non-abbreviated form of hepatocellular carcinoma and then put it in the brackets: “hepatocellular carcinoma (HCC)”, as it is not previously mentioned.
A: Thank you but HCC is spelled out in line 26.
- Line 52, is the word “poor” (or similar) missing next to prognosis? Please check.
A: We added “poor”. Thank you.
- Line 71, 72, 199, 210, 229, 231, 314, 315, 367 please add reference at the end of the sentence.
A: We did.
- Line 72 RFA is already explained, please just mention “RFA”.
A: We corrected.
- Line 73 MWA is already explained, please just mention MWA.
A: We corrected.
- Line 100, please add to the end of the sentence, “...respectively.”
A: We corrected.
- Line 157, it is preferable if the headings are not with abbreviations. Please write heading as “computed tomography high-dose rate brachytherapy (CT-HDRBT)”
A: We corrected and adopted for all procedures.
- Line 161-162, for the term “computed tomography high-dose rate brachytherapy” please only write the abbreviation “CT-HDRBT”.
A: We did.
- Line 167 there is a spelling error, please change “recection” for “resection”.
A: We changed.
- Line 175 please write the full form of “CR”.
A: We did.
- In line 176, it is stated that the “median local tumor control was 10 months”, this was after CT-HDRBT? Explicit please.
A: We did.
- When referring to the number of patients, do so with a number or the number written in words. I suggest doing the numbers, as the majority of the paper is written this way. In line 179 please change “ten” for “10”.
A: We did.
- Paragraph from line 209-215 should be added to the previous one, as the authors are naming three advantages together.
A: We did.
- Line 221 it is preferable if the headings are not with abbreviations. Please write as heading as “Transarterial Embolization (TACE)”
A: We did.
- Line 224 please change “hepatocellular carcinoma” for “HCC” because it is already abbreviated in line 44.
A: We did.
- In line 232 there is a grammatical error, close the parenthesis before the reference.
A: Parenthesis is closed.
- It is suggested that authors use a standard way in mentioning figures. Choose either “Fig number” or (Fig. number), you have used both in the article.
A: It is now consistent.
- Line 236, please add a space in between the number and the cm in “6.5cm”.
A: We did.
- Line 239, eliminate one point after the final sentence, you have two.
A: We did.
- Please remove sentence from line 240 to 243, the section about TACE is extensive and that sentence does not contribute on a large scale.
A: Wie did.
- Please rewrite sentence from lines 243-245.
A: please explain the reason, as we think thematically this sentence makes sense at this point.
- Line 352, eliminate a coma (there are two).
A: We did.
- Line 253 please only write the abbreviation “CR”, because it is previously non-abbreviated in line 175.
A: We did.
- Line 309 please write “Transarterial radioembolization (TARE)”, as it is preferable if the headings are not with abbreviations.
A: We did.
- Line 310, please write only the abbreviation “TARE”.
A: We did.
- In line 323 and 342, please change “ICCA” to “iCCA”.
A: We did.
- Line 360, please change the red dot for a black dot.
A: We did.
Reviewer 3 Report
This manuscript by Auer TA et al reviewed the interventional treatment options for iCCA with intermittently compared with HCC treatment and discussed the potential application of these strategies on combined HCC-CCA. This is a well described review. Given limited treatment options in iCCA and cHCC-CCA, this review provides important references. There are several main flaws need more clarification.
1). The title needs to be modified in order to reflect what the authors discussed in the manuscript and intented to expand. The cholangiocellular carcinoma is only used in the title but completely disappear in the formal context. There is a very limited discussion of cHCC-CCA, though the authors tried to link the study results from mainly from iCCA to cHCC-CCA.
2). It is hard to follow the data from different studies included in this review. It is recommended to have one to two tables to summary the results from different analysis in parallel in order to help readers.
3) based on Nat Rev Gastroenterol Hepatol 2020; 17:557, iCCA is not the most comment type in CCA or BTCs. Please correct accordingly.
Author Response
Reviewer #3:
This manuscript by Auer TA et al reviewed the interventional treatment options for iCCA with intermittently compared with HCC treatment and discussed the potential application of these strategies on combined HCC-CCA. This is a well described review. Given limited treatment options in iCCA and cHCC-CCA, this review provides important references. There are several main flaws need more clarification.
1). The title needs to be modified in order to reflect what the authors discussed in the manuscript and intented to expand. The cholangiocellular carcinoma is only used in the title but completely disappear in the formal context. There is a very limited discussion of cHCC-CCA, though the authors tried to link the study results from mainly from iCCA to cHCC-CCA.
A: Thank you for the comment. As mentioned, there is little data on cHCC-CAA mixied tumors, which is why the interventional therapy options can only be transferred from iCCA to cHCC-CCA. Furthermore according to your advice we changed the title to:
“Interventional Treatment Strategies in intrahepatic Cholangiocarcinoma and Perspectives for Combined Hepatocellular-Cholangiocarcinoma”
2). It is hard to follow the data from different studies included in this review. It is recommended to have one to two tables to summary the results from different analysis in parallel in order to help readers.
A: Since many small studies and case series exist, we do not want to include tables for each procedure. Especially since the data mainly belong to the iCCA, they would confuse the readership, since the transfer to the cHCC-CCA is to be referred to here.
3) based on Nat Rev Gastroenterol Hepatol 2020; 17:557, iCCA is not the most comment type in CCA or BTCs. Please correct accordingly.
A: We did.
Round 2
Reviewer 1 Report
The authors have reasonably addressed the issues raised in my previous review.
Author Response
Reviewer #1:
C1: The authors have reasonably addressed the issues raised in my previous review.
A: Thank you.
Reviewer 2 Report
No more comments
Author Response
Reviewer #2:
C1. No more comments
A: Thank you.
Reviewer 3 Report
Thank you for revision, but:
1) Please provide summarized table for iCCA since the authors mainly discussed about the treatment of iCCA, it will not be confusing.
2) iCCA is second most common in CCA, not in primary liver cancer
Author Response
Reviewer #3:
Thank you for revision, but:
C1: Please provide summarized table for iCCA since the authors mainly discussed about the treatment of iCCA, it will not be confusing.
A: We added two tables. First, for ablational procedures and second for transarterial procedures:
|
Procedures |
Authors |
Design |
Endpoints* |
Outcome |
|
RFA/MWA |
||||
|
RFA |
Fu et al. (2012) |
Monocentric; Retrospective; n=17 |
Tumor necrosis OS (median) RFS (median) 1Y-, 3Y-, 5Y-OS
|
96.2% 33 months 17 months 84.5%; 43.3%; 28.9% |
|
RFA |
Han et al. (2015) |
Meta-Analyses; Systematic Review; n=84
|
1Y-, 3Y-, 5Y-OS Major complications |
82%; 47%; 24% (pooled) 1-2 in 4 studies |
|
RFA/MWA |
Kim et al. (2022) |
Meta-Analyses; Systematic Review; n=917 |
Technical Efficacy 1Y-, 3Y-, 5Y-OS 1Y-, 3Y-RFS 1Y-, 3Y-, 5Y-TTLTP Major complications
|
91.9% 82.4%; 41.1%; 28.5% 40%; 19.2% 79.3%; 59.5%; 58.2% 5.7% |
|
MWA |
Zhang at al. (2018) |
Monocentric; Retrospective; n=107 |
OS (median) 1Y-, 3Y-, 5Y-OS 6m-, 12m-, 18m-, 24m-PFS Major complications
|
28.0 months 93.5%; 39.6%; 7.9% 67.4%; 41.5%; 18.2%; 8.7% 2.8% |
|
MWA vs. SR |
Xu et al. (2019) |
Monocentric; Retrospective; n=121 |
MWA: 5Y-OS (estimated) 3Y-RFS (estimated) Major Complications
Surgical Resection: 5Y-OS (estimated) 3Y-RFS (estimated) Major Complications
|
23.7% 33.1% 5.3% (sign.)
21.8% 30.6% 13.8% (sign.) |
|
MWA + TACE |
Yang et al. (2015) |
Monocentric; Retrospective; n=26 |
OS (median) PFS (median) 6m-, 12m-, 24m-OS Major complications
|
19.2 months 6.2 months 88.5%; 69.2%; 61.5% 0% |
|
Cryoablation |
||||
|
|
Currently no significant evidence available |
|||
|
CT-HDRBT |
||||
|
CT-HDRBT |
Schnaauff et al. (2012) |
Monocentric; Retrospective; n=15
|
OS (median) LTC (median)
|
14 months 11 months
|
|
CT-HDRBT |
Kamphues et al. (2012) |
Monocentric; Retrospective; n=10
|
1Y-, 5Y-OS Major complications
|
77.1%; 51.4% 0% |
|
CT-HDRBT |
Jonczyk et al. (2018) |
Monocentric; Retrospective; n=61
|
iCCA <4cm: OS (median) LTC (median) PFS (median)
iCCA > 4cm: OS (median) LTC (median) PFS (median)
|
15.5 months 8.0 months (sign.) 5.0 months
10.0 months 6.0 months (sign.) 3.0 months |
|
IRE |
||||
|
|
Currently no significant evidence available |
|||
2) iCCA is second most common in CCA, not in primary liver cancer
A: We corrected.